# Genetically Modified Heat Shock Protein90s and Polyamine Oxidases in *Arabidopsis* Reveal Their Interaction under Heat Stress Affecting Polyamine Acetylation, Oxidation and Homeostasis of Reactive Oxygen Species

**DOI:** 10.3390/plants8090323

**Published:** 2019-09-03

**Authors:** Imene Toumi, Marianthi G. Pagoulatou, Theoni Margaritopoulou, Dimitra Milioni, Kalliopi A. Roubelakis-Angelakis

**Affiliations:** 1Department of Biology, University of Crete, Voutes University Campus, 70013 Heraklion, Greece; 2Department of Plant Biotechnology, Agricultural University of Athens, Iera odos 75, 11855 Athens, Greece

**Keywords:** heat shock proteins, heat stress, polyamines, polyamine oxidases, PA acetylation, PA oxidation, PA back-conversion, hydrogen peroxide

## Abstract

**One Sentence Summary:**

Heat shock proteins90 (HSP90s) induce acetylation of polyamines (PAs) and interact with polyamine oxidases (PAOs) affecting oxidation of PAs and hydrogen peroxide (H_2_O_2_) homeostasis in *Arabidopsis thaliana*.

**Abstract:**

The chaperones, heat shock proteins (HSPs), stabilize proteins to minimize proteotoxic stress, especially during heat stress (HS) and polyamine (PA) oxidases (PAOs) participate in the modulation of the cellular homeostasis of PAs and reactive oxygen species (ROS). An interesting interaction of HSP90s and PAOs was revealed in *Arabidopsis thaliana* by using the *pLFY:HSP90RNAi* line against the four *AtHSP90* genes encoding cytosolic proteins, the T-DNA *Athsp90-1* and *Athsp90-4* insertional mutants, the *Atpao3* mutant and pharmacological inhibitors of HSP90s and PAOs. Silencing of all cytosolic *HSP90* genes resulted in several-fold higher levels of soluble spermidine (S-Spd), acetylated Spd (N^8^-acetyl-Spd) and acetylated spermine (N^1^-acetyl-Spm) in the transgenic *Arabidopsis thaliana* leaves. Heat shock induced increase of soluble-PAs (S-PAs) and soluble hydrolyzed-PAs (SH-PAs), especially of SH-Spm, and more importantly of acetylated Spd and Spm. The silencing of *HSP90* genes or pharmacological inhibition of the HSP90 proteins by the specific inhibitor radicicol, under HS stimulatory conditions, resulted in a further increase of PA titers, N^8^-acetyl-Spd and N^1^-acetyl-Spm, and also stimulated the expression of PAO genes. The increased PA titers and PAO enzymatic activity resulted in a profound increase of PAO-derived hydrogen peroxide (H_2_O_2_) levels, which was terminated by the addition of the PAO-specific inhibitor guazatine. Interestingly, the loss-of-function *Atpao3* mutant exhibited increased mRNA levels of selected *AtHSP90* genes. Taken together, the results herein reveal a novel function of HSP90 and suggest that HSP90s and PAOs cross-talk to orchestrate PA acetylation, oxidation, and PA/H_2_O_2_ homeostasis.

## 1. Introduction

Heat stress impairs plant growth and productivity globally [1]. Plants, as sessile organisms, are highly adaptive to harsh environmental conditions and accommodate the developmental programming in a wide temperature range. Survival above optimal temperature conditions is accompanied by a massive accumulation of HSPs [2,3,4]. HSPs maintain and stabilize proteins and transcription factors through heterocomplex formation and folding/unfolding, thus controlling the activation/inhibition of their complex associates [5,6,7,8]. Furthermore, HSPs bind to targeted molecules through specific receptors for translocation to non-peptide target molecules, such as hormones [9,10], and they act in all cellular compartments [11]. 

In *Arabidopsis thaliana*, the *HSP90* gene family consists of seven members [12,13,14,15,16]. Four highly homologous *HSP90* genes (*AtHSP90-1, -2, -3* and *-4*) encode cytosolic proteins, suggesting important functional redundancies [17,18]. They associate with key components of essential cellular processes [19] and play crucial role(s) in areas as diverse as cellular homeostasis, cell growth, development, and organismal evolution [5,20,21,22]. Recent evidence has highlighted the commitment of HSP90 proteins on plant defense mechanisms [23], brassinosteroid or auxin signaling pathways [10,24,25], and stomata patterning [26]. Moreover, HSP90s are essential for the vegetative-to-reproductive phase transition and flower development [22].

The inhibition of HSP90 increases the stochastic variation inherent to developmental processes by releasing morphogenic traits controlled by the HSP90 buffering capacity of mutations. Administration of radicicol (specific HSP90 inhibitor) to wild type (WT) plants reduces viability under physiological conditions [5], whereas many homozygous *hsp90* mutant lines are lethal [15,27]. In *Arabidopsis*, the *Athsp90c-1* mutant exhibits a high rate of albinos and aborted seeds, confirming that the HSP90 protein localized specifically to chloroplasts is essential for viability [27]. HSP90s interact with tetrapyrroles, by modulating the photosynthesis-associated nuclear genes (*PhANG*) expression, in response to oxidative stress [28]. They associate with the suppressor of the G2 allele of *skp1* (*SGT1*) for the stabilization of the nucleotide-binding domain and leucine-rich repeat-containing (NLR) immune sensors, which mediate plant defense mechanisms [29]. In addition, HSP90s participate in stress signaling pathways [18,30,31,32,33]. In fact, under stress, the inhibition of HSP90 induces, in most cases, deregulation of the putative client’s functionality and alteration of cell processes [31,34].

Polyamines are highly reactive aliphatic polycations [35]. The more abundant and best studied PAs are the diamine putrescine (Put), the triamine Spd and the tetramine Spm. Polyamine homeostasis is determined by a complex regulatory mechanism which includes turnover, export/transport, as well as modifications of their amino-groups, *via* mechanisms such as acetylation and conjugation [36,37,38,39]. Polyamines, *per se* or *via* their metabolic products such as H_2_O_2_, interfere with a plethora of dynamic metabolic and developmental processes [40,41,42], as well as stress responses [37,38,43,44,45,46,47]. Also, PAs are highly involved in adaptative mechanisms of hyperthermophilic proteins [48], whereas they interfere with HSP synthesis under increasing temperatures by an unknown mechanism [49].

Polyamine oxidases participate in the regulation of PA homeostasis, mediating their oxidation/back-conversion [39,50]. In *Arabidopsis thaliana*, oxidation/back-conversion of PAs is mediated by the peroxisomal *AtPAO2* (At2g43020), *AtPAO3* (At3g59050), *AtPAO4* (At1g65840) and the cytosolic *AtPAO1* (At5g13700) and *AtPAO5* (At4g29720) [46]. Products of PAO-mediated enzymatic action include aldehydes and H_2_O_2_ [43,47]. Recently, PAOs were confirmed to play an important role in the control of cell proliferation (in animals) through the generated toxic aldehydes and H_2_O_2_ and were proposed as fine competitors for antiproliferative therapies [51]. In plants, the generated H_2_O_2_, depending on its "signature", can signal either the orchestration of tolerance to abiotic/biotic stresses or execution of programmed cell death (PCD) [37,38,39,52,53,54,55,56,57,58,59,60]. Furthermore, an NADPH-oxidase/PAO feedback loop controls oxidative burst under salinity in tobacco [61]. In animals [62], but not in plants [47,63], acetylated PAs are the major form of the conjugated PAs and the favorite substrate for PAOs.

Evidence for the interplay between HSPs and PAs was first provided for the HSP70 family. Depletion of PAs in a cell line derived from rat hepatoma induces direct inhibition of *HSP70* expression [64]. In addition, exogenous Spm increases the transcription of *HSP90* and subsequently enhances heat tolerance in *Arabidopsis thaliana* [65]. Furthermore, the underexpression of PAO results in thermotolerant tobacco plants [66]. Such findings suggest the existence of an evolutionary conserved network which acts for co-regulation of PAs and HSPs to efficiently orchestrate HS response. However, the exact mechanism remains elusive. In this work, we attempted to identify potential links between *HSP90* and PA homeostasis implicating more specifically *PAO* genes, PAO activity and H_2_O_2_ levels. Impaired *HSP90* action, derived either from the *HSP90RNAi* effect or from pharmacological inhibition of the HSP90 protein activity, induced increase in PA titers, especially of the acetylated-PAs, N^8^-acetyl-Spd and N^1^-acetyl-Spm and the conjugated forms. The interconnection between HSP90 and PAs was further determined since loss of PA homeostasis by impairing a major path of PA oxidation had a specific impact on *HSP90* both at transcriptional and translational levels. Similarly, genetic modification of HSP90 had a profound effect on genes involved in the PA oxidation pathway. Our results reveal a novel interaction by which *HSP90*s/*PAO*s co-regulate the intracellular titers of PAs/H_2_O_2_ with an unknown yet impact on stress response. 

## 2. Results

### 2.1. Underexpression of AtHSP90 1–4 Genes Results in Higher S-PAs and S-N^8^/N^1^ Acetylated PA Titers

Polyamine homeostasis and *PAO* expression/PAO activity are cell/tissue/organ- specific, strongly affected by the ontogenetic stage of the plant/organ and the growing conditions [40,41]. In order to identify any potential link between HSP90 and PA homeostasis, the endogenous PA titers were analyzed in the leaves of 15 day-old WT, *Athsp90-1* and *Athsp90-4* mutants and *pLFY:HSP90RNAi* transgenic plants. More specifically, the titers of soluble Put (S-Put), S-Spd, S-Spm, and the two mono-acetylated PAs, N^8^-acetyl-Spd and N^1^-acetyl-Spm were determined (Figure 1A). The leaves of *Athsp90 1–4* contained significantly higher levels of S-Spm and acetyl-PAs and the *pLFY:HSP90RNAi* line contained higher levels of S-Put, S-Spd and S-Spm compared to WT. Interestingly, the *pLFY:HSP90RNAi* transgenics exhibited a nearly 18-fold increase of (N^8^-acetyl-Spd + N^1^-acetyl-Spm) compared to the WT (Figure 1A). 

To further confirm that genetic inhibition of the *HSP90* genes affects PA homeostasis independently of plant age, PA contents were determined in 35 day-old mutants/transgenics and WT plants. Consistently, although a general decrease in PA levels was observed compared to the 15 day-old seedlings, in accordance with our previous results for tobacco and grapevine [40], all three lines, *Athsp90 1–4*, and *pLFY:HSP90RNAi* contained significantly greater S-Spm with the highest levels recorded in the *pLFY:HSP90RNAi* line, which also contained higher S-Put (Figure 1B). The most striking effect of *AtHSP90* deregulation on PA titers was, as in the 15 day-old plants, the significant increase of S-acetylated forms of Spd (N^8^-acetyl-Spd) and Spm (N^1^-acetyl-Spm) in the *Athsp90* transgenic lines (Figure 1B). The picture was somehow different in the conjugated SH-fraction (Appendix A). The *Athsp90 1–4* lines contained significantly higher SH-Put, SH-Spd and SH-Spm, as well as SH-acetyl-PAs, when compared to the corresponding titers in the S-fraction whereas the *pLFY:HSP90RNAi* line contained significantly lower amounts of SH- and acetyl-PAs, when compared to the *Athsp90 1–4* mutants and to WT (Appendix A). The results point strongly to a complex mechanism connecting *HSP90* and PA homeostasis which may be ontogenetically regulated.

### 2.2. The Increased PAs in the Athsp90 Mutants Correlate with Increased H_2_O_2_ Content. HSP90s Modulate Free Radical Production

Under normal conditions, PA back-conversion catalyzed by PAOs leads to H_2_O_2_ production balancing the intracellular oxygen consumption rate [38]. In order to examine whether HSP90s modulate H_2_O_2_ production, the in situ semi-quantitative method based on staining with 3,3’-diaminobenzidine (DAB) was employed. We found that all 15 day-old *Athsp90* and *pLFY:HSP90RNAi* seedlings had increased H_2_O_2_ content compared to WT seedlings, as assessed by DAB staining (Figure 2). Hydrogen peroxide levels paralleled the increased S-PAs titers, which are substrates of PAOs; the *pLFY:HSP90RNAi* line as well as the *Athsp90-1* and the *Athsp90-4* young seedlings, which contained the highest S- and acetyl-PAs, exhibited the highest H_2_O_2_ levels compared to WT (Figure 1A and Figure 2). These results further point to a certain correlation between *HSP90*s, PAs, and PAO-generated H_2_O_2_.

### 2.3. Under HS, Genetic Depletion and Pharmacological Inhibition of HSP90 Affect PA Homeostasis and Enhance Acetylated Forms 

Since HS induces transcription of the *HSP90* genes [12,13,16], it was of interest to assess whether genetic depletion of *HSP90 1–4* genes by RNAi [22] or pharmacological inhibition of HSP90 in WT *Arabidopsis* plants would similarly affect PA homeostasis under HS conditions. Thus, analysis of PA titers was performed in leaves of WT and *pLFY:HSP90RNAi* transgenic plants following acute HS (1 h at 42 °C) in the presence (in WT) or absence (in *pLFY:HSP90RNAi*) of radicicol (Rad), a specific inhibitor of HSP90 activity. In WT, HS resulted in increased S-Spm, and considerable decrease of SH-Spd and SH-Spm levels (Figure 3). Soluble-Spm is a common thermo-responsive indicator and has been proposed as a marker of thermotolerance [65]. The combined effect of HS + Rad resulted in increased S-Put, S-Spd, SH-Put, SH-Spd and SH-Spm when compared to HS treatment alone. In the WT, the S- N^8^-acetyl-Spd and N^1^-acetyl-Spm titers were slightly affected by HS, whereas the inhibition of HSP90 by Rad resulted in significant increase of the N-acetyl-PAs in both fractions S and SH (Figure 3). Similarly, genetic depletion of cytosolic HSP90s in the *pLFY:HSP90RNAi* line resulted in a significant increase of all S-PAs, SH-PAs as well as of S-acetyl-PAs and SH-acetyl-PAs under HS. In particular, S-Spd in *pLFY:HSP90RNAi* showed a nearly 3-fold increase (Figure 3). These data indicate that impairment of the HSP90 molecular chaperones, when coupled to the stimulatory effect of the HS-related pathway, increases further the deregulation of PA homeostasis and leads to a significant increase of acetylated PAs.

### 2.4. Genetic and Pharmacological Inhibitions of HSP90 Affect the PA Oxidation Pathway

Taking into consideration the changes in PA titers and the high levels of H_2_O_2_ in the RNAi *pLFY:HSP90RNAi* line (Figure 1 and Figure 2), we studied the expression profile of *AtPAO1*, *3*, *5* and *AtHSP90 1–4* genes in WT and this transgenic line. To explore the impact that *HSP90* could have on PA homeostasis, we studied the expression of *AtPAO1*, *AtPAO3* and *AtPAO5* genes in *pLFY:HSP90RNAi* plants in which cytosolic HSP90 proteins are markedly depleted [16,20]. In WT plants, HS increased mRNA levels of *AtHSP90-*1 and *AtHSP90-4*; treatment with HS + Rad also induced high *AtHSP90-1* and *AtHSP90-4* transcript levels when compared to the controls. It is worth noticing that the abundance of *AtHSP90-1* is slightly lower when HS coupled with Rad than HS alone (Figure 4A). 

*AtPAO* mRNAs in the *pLFY:HSP90RNAi* lines were increased when compared to WT under control conditions, which was not remarkably different under HS (Figure 4B). If the expression of *PAO* genes was affected by HSP90 function, then Rad should modulate gene transcription pattern similarly to genetically compromised HSP90 activity. In fact, Rad application on WT plants induced the expression of the *AtPAO1*, *AtPAO3* and *AtPAO5* genes (Figure 4B). Consequently, pharmacological HSP90 inactivation affected the expression of *PAO* genes in a similar way to *HSP90* mRNA depletion by RNAi silencing. Similar results were obtained when the abundance of PAO3 protein was assessed. The levels of PAO3 protein correlated well with the corresponding mRNA levels in both the WT and the *pLFY:HSP90RNAi* plants under the different treatments (Figure 4C). 

### 2.5. Inhibition of HSP90 under HS Induces Expression of NATA1, a Putative Acetyltransferase-Like Gene 

As shown above, leaves of *pLFY:HSP90RNAi* plants exposed to HS as well as leaves of WT exposed to HS coupled with Rad, exhibit a noticeable increase of acetylated forms of higher PAs (Figure 3). The *Arabidopsis*
*NATA* genes are the closest homologues of mammalian PA acetyltransferases. Thus, due to the lack of identified genes regulating the acetylation of Spd and Spm in *Arabidopsis*, we prompted to examine the transcript levels of *NATA1* gene (At2g39030) which is identified as a Put and ornithine acetyltransferase [67,68], as a means to assess the genetic trend of PA acetylation under our conditions. Again, and by using excised leaves incubated in Morpholino Ethane Sulfonic acid (MES) and exposed to HS with (WT)/without Rad (*Athsp90s*), the WT and the *PLFY:HSP90RNAi* mutant under control conditions contained low amounts of *NATA1* mRNA. Heat shock induced *NATA1* expression in the WT and *PLFY:HSP90RNAi* plants, exhibiting a 2-fold increase of the transcript levels. Interestingly, in WT plants *NATA1* transcripts increased more than 5-fold under HS in the presence of Rad. The results show that inhibition of HSP90 by the HSP inhibitor Rad releases PA acetylation process and suggest that PA acetylation is in fact inhibited by HSP90 (Figure 4B,C).

### 2.6. PAO-Induced Stimulation during HSP90 Inhibition Is Responsible for the Increased H_2_O_2_


Since inhibition of HSP90 induced *PAO* transcription and increased PAO protein levels (Figure 4B,C), supporting cross-talk between the inhibition of HSP90 and the stimulation of the PAO pathway, we were prompted to further establish this relationship; thus, PAO activity was determined, as H_2_O_2_ production, in leaves of WT plants treated with HSP90 inhibitor and in *PLFY:HSP90RNAi*, under HS. The results confirmed the activation of PAO under such conditions (Figure 5A and Appendix A). To further test this finding, we also determined PAO activity *in planta* intact WT leaves infiltrated with Rad or Spd (as a positive control) and submitted to HS (Figure 5B). In addition, the results were confirmed by determining the endogenous H_2_O_2_ of leaves treated as above, which confirmed that in fact high PAO activity correlates with high H_2_O_2_ levels (Figure 5C). Finally, the increased amounts of H_2_O_2_ were further verified by in situ staining of WT leaves under HS and HS + Rad. Consistently, guazatine (specific PAO inhibitor), when administered simultaneously with Rad, reduced H_2_O_2_ levels, supporting that the higher PAO activity contributes to H_2_O_2_ generation upon HSP90 inhibition (Figure 5D). As with WT, the use of guazatine along with HS attenuated PAO activity in *PLFY:HSP90RNAi* leaves (Appendix A).

### 2.7. PAO and AtHSP90 Reciprocally Affect Each Other’s Expression

To examine whether PAO homeostasis has an impact on *AtHSP90* expression, *Atpao3* loss-of-function plants were tested. By applying HS to leaves incubated in MES medium, transcript analysis revealed that only *AtHSP90-1* and *AtHSP90-4* mRNAs were increased in the *Atpao3* mutant compared to WT. The abundance of *AtHSP90 1–4* mRNAs as well as HSP90 protein were analyzed in the *Atpao3* mutant and the WT, under HS (Figure 6A,B). In the *Atpao3*, mRNA titers of the *AtHSP90-1* and *AtHSP90-4* were significantly greater compared to WT, whilst HS resulted in an obvious increase of the mRNA levels of the *AtHSP90 1–4* genes (Figure 6A). Inhibition of HSP90 activity simultaneously to HS stimulation (HS + Rad) exerted no further significant effect on the *AtHSP90 1–4* mRNA levels (Figure 6A). 

It is known that HSP90 are essential molecular chaperones in eukaryotic cells, with key functions in signal transduction networks. To test whether HSP90 homeostasis is disrupted when the PA oxidation pathway is challenged, the molecular chaperone protein abundance was assessed. Immunoreactive HSP90 protein levels exhibited parallel increase under HS in both, WT and *Atpao3* and was more abundant in the *Atpao3* genotype (Figure 6B and Appendix A). In *pao3* mutant, the HSP90 protein levels were increased.

Furthermore, semi-quantitative reverse transcriptase (RT)-PCR revealed that the cytosolic *AtHSP90 1–4* genes in *Atpao3* seedlings (5, 10, and 15 day-old) revealed a general over-expression of these genes in the *Atpao3* mutant compared to WT, at different time points (Appendix A).

Interestingly, the *Atpao3* leaves contained lower endogenous mRNA levels of all four *AtPAO* genes (*AtPAO1*, 2, 4 and 5). Pharmacological inhibition of HSP90 activity under HS (HS + Rad) induced an increase in the *AtPAO* transcripts and restored the *AtPAO* mRNAs to levels similar to the WT control, except the *AtPAO4* that remained low (Appendix A). The restoration of the *AtPAO* mRNA levels in the *Atpao3* mutant subsequently to the inhibitory effect of Rad on HSP90 activity suggests that *AtPAOs* are activated subsequently to *HSP90* deregulation. Application of HS + Rad did not induce accumulation of acetyl-S-PAs in the *Atpao3* mutant genotype (Appendix A).

## 3. Discussion

The hypothesis of cross-talk between HSPs and PAs was initially evoked through the finding that exogenous Spm increases mRNA levels of *HSP90* and subsequently enhances heat tolerance in *Arabidopsis* [65]. Results herein support this hypothesis and reveal reciprocal molecular and biochemical interactions of *HSP90*/*PAO* genes and/or HSP90/PAO proteins. Depletion of cytoplasmic HSP90 results in increased levels of PAs and more specifically of N^8^-acetyl-Spd and N^1^-acetyl-Spm, which, under physiological conditions, are present only in traces in the leaves of WT (Figure 1) [69]. In animals, Spd/Spm acetyltransferase (SSAT) is a key enzyme participating in the first step of PA catabolism, mediating the acetylation of PAs whereas PAOs catalyze the final oxidation of acetyl-PAs. Overexpression of *SSAT* correlates with growth inhibition in a wide range of tumors [70,71,72]. Accumulation of acetyl-PAs by the increased activity of SSAT shifts the metabolism of PAs into lipids and carbohydrates through the increasing consumption of acetyl-CoA and ATP, leading to generation of ROS and, subsequently, to oxidative stress and proteotoxicity [71]. On the contrary, the acetyl-PAs are not the preferred substrates for PAOs in plants [39,55]. Acetylated-Put is directly linked to the pathogen-defense related signals [67] as well as the root cell de-differentiation process, probably as transient metabolite for GABA production [73]. The factors controlling the tri- and tetraamine (Spd and Spm) acetylation remain obscure. *NATA1* is one of the few genes participating in the N-acetylation of amine metabolism in *Arabidopsis*; ornithine is the favorite substrate when stimulated *via* methyl jasmonate [67] and Put as specific substrate when stimulated through jasmonate and salicylic acid [68], both involving the pathogen infection process. Analysis of the mRNA levels of the *NATA1* in *pLFY:HSP90RNAi* HS-treated leaves versus WT treated with HS +/- Rad confirms the stimulation of the *NATA1* transcription under HS in both genotypes, and a further stimulation when HS is associated with HSP90 inhibitor in the WT (Figure 4B). 

Both, HSP90s and PAs, are highly responsive to environmental challenges. HSP90s increase under stress, including heat, osmotic, and heavy metals, as well as biotic stress, whereas PAs exhibit complex profiles under such conditions [1,12,32,37,38,74,75,76]. Spermine plays a significant role in *Arabidopsis* thermotolerance, and exogenously applied Spm enhances HS response by inducing increase of mRNA levels of *HSP* genes [65]. Although sufficient information on the implication of HSP in proteotoxicity is lacking, it is well established that HSP chaperones are highly responsible for the prevention of protein degradation and misfolding and for preventing non-native proteins from non-specific intermolecular interactions, eliminating proteotoxicity propagation [77,78]. Nishizawa-Yokoi et al. [79] propose a model for HSP90s role in ROS-induced proteotoxic stress in *Arabidopsis thaliana*. They suggest that ROS accumulation results in inhibition of proteasome 26S, which induces a peculiar poly-ubiquitination. HSP90s, which act as Heat Shock Factor (HSF) repressors under normal conditions, release the HSF which regulates in turn the expression of a Heat Shock Transcription Factor A2 (*HSFA2*). Also, HSP90s regulate the production of superoxides mediated by NADPH-oxidase but are not required for the generation of H_2_O_2_ in human embryonic kidney cells [75]. Pharmacological inhibition of HSP90 in PC-12 cells induces cytotoxicity and cell death *via* oxidative stress by an unidentified mechanism [34]. Taking into consideration the involvement of HSP90 in cell proliferation processes, we can speculate that both, HSP90s and PAs may be integrated in a specific biochemical pathway through the competitive regulation of cell cytotoxicity and ROS generation involving potentially the homeostatic regulation of oxidation and/or acetylation of PAs. 

The drastic reduction of HSP90s results in deregulation of PA titers and induction of the PAO pathway, which generates H_2_O_2_. In WT, the pharmacological inhibition of HSP90s under HS induces expression of *AtPAO* genes, suggesting that HSP90s may be implicated in a way to decelerate the action of PAOs into the cell. Indeed, the *pLFY:HSP90RNAi* line exhibits elevated endogenous *AtPAO* mRNA levels similar to those in WT when treated with HS + Rad, suggesting that biochemical inhibition of HSP90s defines a favorable endogenous potential of PA oxidation (Figure 4 and Figure 5).

As key agents in the regulation of thermotolerance, HSP90s show strong connection with other stress-related factors in signal perception and transduction [18,80]. Within this context, HS induces in both WT and *pLFY:HSP90RNAi* mutant an increase of H_2_O_2_, which is a predictable trait of stress (Figure 5C). Simultaneous application of HS + Rad enhances H_2_O_2_ generation in WT. In parallel with the induction of PA oxidation, the use of guazatine, a specific inhibitor of PAO, attenuates the HS + Rad-dependent accumulation of H_2_O_2_, confirming the concomitant link between HSP90 inhibition/PA oxidation-back-conversion/H_2_O_2_ generation. HSP90 inhibition results in *AtPAOs* expression and increased transcript levels, especially the *AtPAO1* and *AtPAO5*. Within this context and the specific affinity of the two proteins AtPAO1 and AtPAO5 to Spm (Nor and especially T-Spm) [52,81], we stress the specific implication of Spm oxidation in HS response. *AtPAO3* is also highly affected by the deletion of the HSP90 action but seems to follow different pattern. Furthermore, the transcriptional analysis of the four cytosolic *AtHSP90* genes in the *Atpao3* mutant reveals the high endogenous *AtHSP90* transcript levels in this mutant and once again the putative mutual inhibitory effect.

Overall, our results suggest a link between HSP90s and PAs/PAOs/H_2_O_2_ under HS. As schematically presented in the proposed model (Figure 7), HSP90 proteins participate in the regulation of PA conjugation, more specifically, in PA acetylation. The *pLFY:HSP90RNAi* plants exhibit: (i) overexpression of the major *AtPAO* genes and consequently increased PAO protein and activity levels, which result in increased H_2_O_2_ intrinsic levels; and (ii) significant increase of PAs and, especially, of acetylated higher PAs titers. In turn, underexpression of the *AtPAO3* gene (*Atpao3* mutant), induces overexpression of one or more of the *AtHSP90 1–4* genes. Under conditions of HS as pivotal HSP90 stimulator: (1) the transgenic reduction of the *HSP90* genes *(pLFY:HSP90RNAi)*, as well as the pharmacological inhibition of the HSP90 proteins *via* Rad affect similarly both pathways. Namely: (i) positive regulation of the *AtPAOs* and further increase of the PAOs-generated H_2_O_2_, which is also observed under HS conditions in the WT, but to a lesser extent, suggesting that HSP90s are involved in the regulation of the stress-induced PA-oxidation/back-conversion pathway; and, (ii) increase of PAs and acetylated-higher PAs, showing a specific heat-related HSP90 pathway as revealed by HSP90 inhibition. Consequently, by being powerful modulators of PAOs, HSP90s affect the PAO-generated H_2_O_2_, which participates in the stress-induced downstream signaling cascade that is tidily related to ROS homeostasis and PCD [38,43,50,51,61,82,83].

Taken all together, these results reveal a mutually antagonistic HS-regulated effect involving HSP90s, PAs, PAOs and H_2_O_2_ in *Arabidopsis thaliana*.

## 4. Materials and Methods

### 4.1. Plant Material and Treatments

*Arabidopsis thaliana* (ecotype Col-0, WT), an *pLFY:HSP90RNAi* silencing line against the four cytosolic *AtHSP90* genes [22], two T-DNA insertion mutant lines for the *Athsp90-1* and *Athsp90-4* (SALK_007614 and SALK_084059, respectively) and a T-DNA loss-of-function *Atpao3* (SALK_121288) line were used. In vitro produced seedlings were germinated on ½ MS medium [84] supplemented with 0.5% sucrose on Petri dishes and plants were cultured in pots in a controlled growth chamber (23 °C and 8/16 h photoperiod under 150 μmol m^−2^ s^−1^). Fully expanded leaves were excised from 4-week-old vegetative plants, before the floral transition phase, incubated in MES medium (pH 5.7) with agitation for 24 h, and subjected to different treatments: controls were incubated at 23 °C and HS samples at 42 °C, in the presence or absence of 10 mM Spd, 50 µM guazatine (specific inhibitor of PAO), 300 nM radicicol (specific inhibitor of HSP90) or 10 mM H_2_O_2_. Each chemical was added to the medium accompanied by brief vacuum treatment, 10 min prior to HS. All experiments were performed twice in identical growth chambers and growth conditions, with 4 replicates per treatment randomly designed. Samples were collected after 1 h treatment. Eight replicates from two independent experiments were pooled for statistical analysis. Asterisks indicate statistically significant differences from the WT control (*p* < 0.05).

### 4.2. Construction of HSP90-RNAi Line (pLFY:HSP90RNAi Mutant) 

The construction of the *HSP90-RNAi* line is described in [22]. In this work, we analyzed the *pLFY:HSP90RNAi* mutant, line 10.1

### 4.3. RNA Extraction, RT-PCR and Semi-Q RT-PCR Analysis

Total RNA was extracted using the TRIZOL method (modified from Chomczinsky and Sacchi, [85]), any DNA contamination was removed with RQ1 RNase free DNase (Promega, Madison, WI, USA). About 1 μg was used as template in first-strand cDNA synthesis using PrimeScriptTM 1st Strand cDNA Synthesis Kit (TaKaRa, Kusatsu, Shiga, Japan), according to the manufacturer’s instructions. RT-PCR Transcript analysis was performed using the gene specific primers, sequences relative sizes for RT-PCR are: *AtPAO1*: 468 bp, *AtPAO3*: 624 bp, *AtPAO5*: 269 bp, *AtHSP90-1*: 725 bp, *AtHSP90-2*: 197 bp, *AtHSP90-3*: 352 bp and *AtHSP90-4*: 234 bp (Appendix A).

For the semi-Q RT-PCR, 5, 10 and 15 day-old seedlings of *Arabidopsis* wild type (Col) and *Atpao3* were used. First strand cDNA was synthesized using total RNA from each plant sample using SuperScript II Reverse Transcriptase (Invitrogen) in a 20 μL reaction. PCR amplification for each transcript (*AtHSP90-1*: 392 bp, *AtHSP90-2*: 192 bp, *AtHSP90-3*: 2531 bp and the *AtHSP90-4*: 128 bp) was performed using the gene-specific primers (Appendix A). RT-PCR and semi-Q RT-PCR reactions were normalized using the *Ubq* and *18S r*DNA *Arabidopsis* genes, respectively [86]. For all semi-Q RT-PCRs, three technical replicates were prepared from materials pooled from all biological replicates/experiments. At least three variations in cycle number were used to verify reproducibility and amplification in the logarithmic phase, respectively. To estimate the amount of PCR product obtained, PCR reactions were run on ethidium bromide-containing agarose gels and subsequently analyzed in UV light. During gel analysis and data collection, volumes and settings were chosen to avoid saturation of fluorescence signals. RT and semi-Q RT-PCR products were quantified by measuring the intensity per unit area with Image J. The gene specificity of RT-PCR products was confirmed by sequencing. Data represent the mean of the technical replicates.

### 4.4. Determination and In Situ Localization of Hydrogen Peroxide

Hydrogen peroxide was quantified *via* a chemiluminescence assay [87]. In situ H_2_O_2_ was detected by DAB staining using the method of Thordal et al. [88].

### 4.5. Extraction and Quantification of PAs

Polyamines were extracted and analyzed according to Kotzabasis et al. [89], using an HP 1100 high performance liquid chromatographer (HPLC; Hewlett-Packard, Wadbronn, Germany). 

### 4.6. Protein Extraction and Enzyme Assays

Total proteins were extracted as described in Papadakis and Roubelakis-Angelakis [90]. Protein content was measured according to the Lowry method [91]. PAO activity was measured according to the method of Yoda et al. [43] with minor modifications; fifty μg protein extract were incubated with 10 mM Spd for 30 min and measurement of generated H_2_O_2_ was performed for 1 min using a luminometer in a Tris-HCl (pH 8.0) reaction buffer containing 125 mM luminol [92].

### 4.7. Western Blotting

Protein extracts were prepared with Laemmli buffer and 50 μg of denaturated protein aliquots were loaded on 10% (*v*/*v*) polyacrylamide gel and electroblotted onto nitrocellulose membrane. Target proteins were then immunodetected against specific antibodies: a rabbit polyclonal anti-AtPAO3 anti-serum at a 1:10,000 dilution [39] and a monoclonal mouse anti-HSP90 at a 1:2000 dilution (Anti-HSP90 Mouse mAb, AC88, Calbiochem). For anti-AtPAO3 and anti-AtHSP90 horseradish peroxidase-conjugated anti-rabbit IgG (1:5,000) (Sigma, St Louis, MI, USA), or anti mouse IgG (1:10,000) (Sigma, St Louis, MI, USA) were used respectively. Western blots were developed using the Amersham ECL Prime Western Blotting Detection Reagent Kit (Pittsburg, CA, USA).

### 4.8. Image and Statistical Analysis

Image analysis was performed using ImageJ v 1.41 software, and statistical analysis using one-way ANOVA and the Duncan’s test in the STATISTICA software. 

## Figures and Tables

**Figure 1 plants-08-00323-f001:**
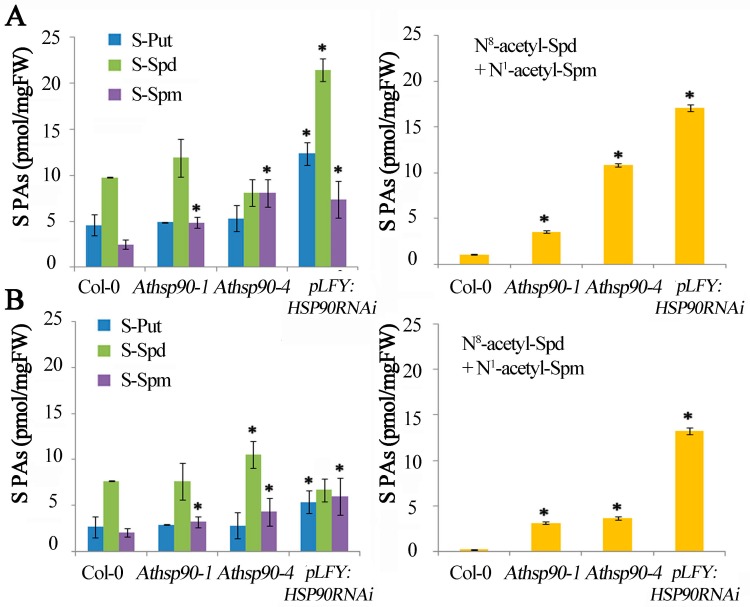
Endogenous polyamine titers in WT, *Athsp90-1*, *Athsp90-4* and *pLFY:HSP90RNAi* mutants of *Arabidopsis thaliana*. (**A**) Soluble Putrescine, Spermidine and Spermine (S-Put, S-Spd, S-Spm) and Soluble acetylated (N^8^-acetyl-Spd + N^1^-acetyl-Spm) polyamine contents in leaves of 15 day-old seedlings. (**B**) Soluble (S-Put, S-Spd, S-Spm) and soluble acetylated (N^8^-acetyl-Spd + N^1^-acetyl-Spm) polyamine contents in leaves of 35 day-old WT, *Athsp90-1*, *Athsp90-4* and *pLFY:HSP90RNAi* plants. Values from 8 replicates from two independent experiments were pooled for statistical analysis. Asterisks indicate statistically significant differences from WT control (*p* < 0.05).

**Figure 2 plants-08-00323-f002:**
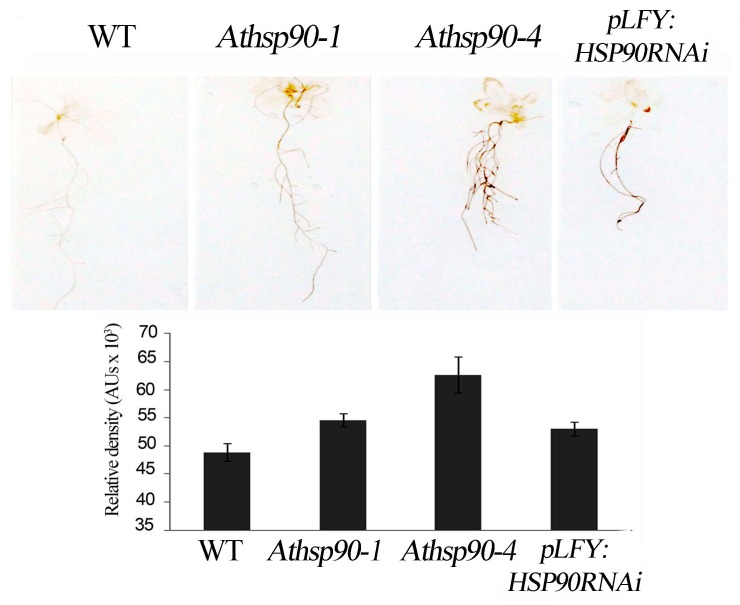
In situ DAB detection of hydrogen peroxide in WT, *Athsp90-1*, *Athsp90-4* and *pLFY:HSP90RNAi* seedlings of *Arabidopsis thaliana* grown on Murashige and Skoog (MS) plates and relative pixels density of leaf staining. Values from 8 replicates from two independent experiments were pooled for statistical analysis.

**Figure 3 plants-08-00323-f003:**
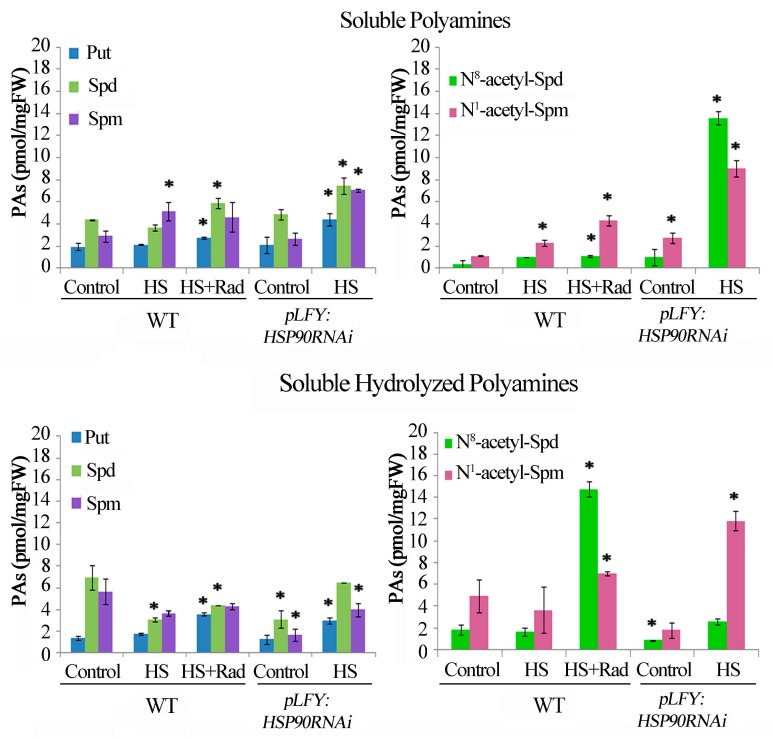
HSP90 activation/inhibition affects polyamine homeostasis in *Arabidopsis thaliana*. Endogenous Soluble (Put, Spd, Spm); Soluble acetylated (N^8^-acetyl-Spd and N^1^-acetyl-Spm); Soluble hydrolyzed (Put, Spd, Spm); and Soluble hydrolyzed acetylated (N^8^-acetyl-Spd and N^1^-acetyl-Spm) polyamine contents following HSP90 activation/inhibition in treated leaves. Leaves of WT and *pLFY:HSP90RNAi* transgenic plants followed brief HS (1 h at 42 °C) in the presence (in WT) or absence (in *pLFY:HSP90RNAi*) of radicicol (Rad). Values from 8 replicates from two independent experiments were pooled for statistical analysis. Asterisks indicate statistically significant differences from the WT control (*p* < 0.05).

**Figure 4 plants-08-00323-f004:**
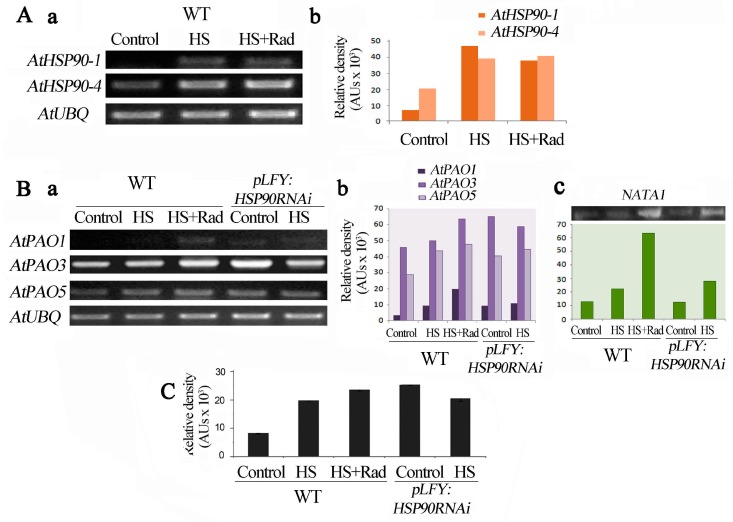
Inhibition of HSP90 affects levels of mRNAs of *AtPAO1*, *AtPAO3*, *AtPAO5* and *NATA1* genes, and PAO3 protein in *Arabidopsis thaliana*. (**A**) a. Abundance of transcripts of *AtHSP90-1* and *AtHSP90-4* genes in WT, b. relative pixels density of gel bands’ intensities following normalization with *AtUBQ*. (**B**) a. Abundance of transcripts of *AtPAO1*, *AtPAO3* and *AtPAO5*, b. relative pixels density of gel bands’ intensities following normalization with *AtUBQ*, c. *NATA1* mRNA levels in WT and *pLFY:HSP90RNAi* mutant, with relative densitometric measurements following normalization with *AtUBQ*. (**C**) PAO3 immunoreactive protein relative levels in the leaves under control, HS and HS + Rad conditions. Image software was used for numeric determination and quantification of gels bands’ intensities.

**Figure 5 plants-08-00323-f005:**
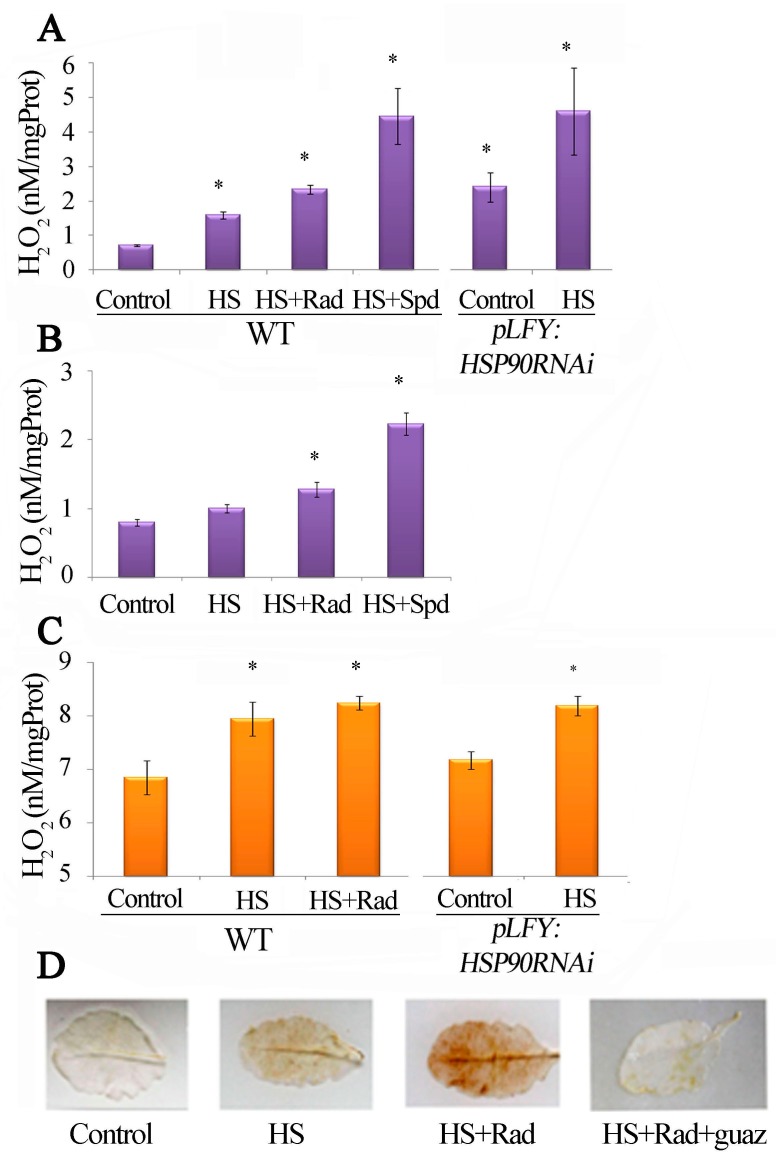
Inhibition of HSP90 stimulates the polyamine oxidation activity. Polyamine Oxidase (PAO) activity measured and evaluated through hydrogen peroxide production in: (**A**) Excised WT leaves cultured on MES medium and exposed to HS with/without Rad; and excised *pLFY:HSP90RNAi* line leaves cultured on MES medium and exposed to HS. (**B**) *In planta* WT leaves infiltrated separately with water (control or HS); Rad, and Spd (as positive control) prior to exposure to HS. (**C**) Luminometric hydrogen peroxide quantification in WT leaves exposed to HS and HS + Rad, and *pLFY:HSP90RNAi* mutant exposed to HS. (**D**) In situ DAB staining of hydrogen peroxide in WT control leaves treated with HS and HS + Rad and with HS + Rad + guazatine (Guaz). Values from 8 replicates from two independent experiments were pooled for statistical analysis. Asterisks indicate statistically significant differences from WT control (*p* < 0.05).

**Figure 6 plants-08-00323-f006:**
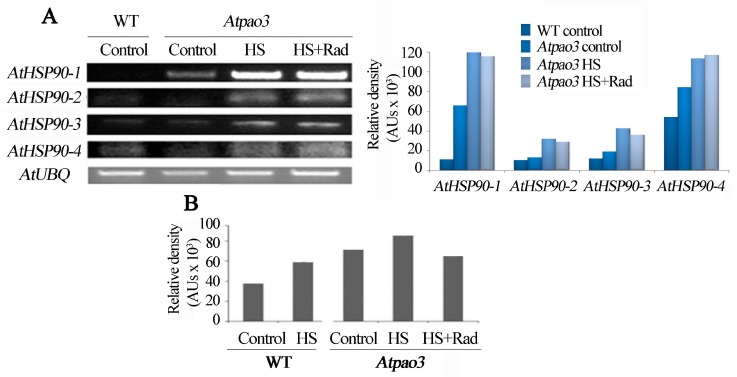
Abundance of cytosolic *AtHSP90* mRNA in WT and *Atpao3* mutant and HSP90 protein levels. Leaves were treated with HS and HS + Rad for 1 h at 42 °C. (**A**) Reverse transcriptase (RT)-PCR of the four *AtHSP90* cytosolic genes (*AtHSP90 1–4*) and relative intensities of bands normalized against ubiquitin. (**B**) Quantification of the HSP90 immunoreactive protein in control and HS- treated WT leaves, as well as in controls, HS and HS + Rad treated *Atpao3* leaves.

**Figure 7 plants-08-00323-f007:**
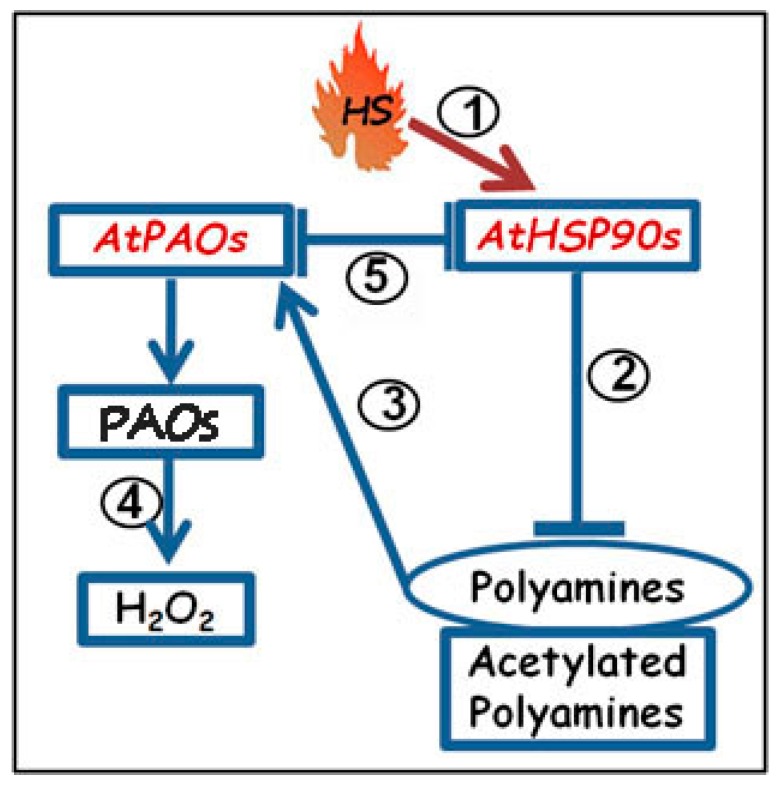
Proposed model of Heat Shock Protein90s (HSP90s)/Polyamine oxidases (PAOs)/Polyamines (PAs)/H_2_O_2_ cross-talk in *Arabidopsis thaliana*: Under conditions of HS, as pivotal HSP stimulator (1), the transgenic/pharmacological reduction of *HSP90* genes/HS proteins stimulate an increase of PA and acetylated-PA levels (2), a positive regulation of *AtPAO*s (3), and further increase of the PAO-related H_2_O_2_ (4). Reciprocally, underexpression of *AtPAO* in *Atpao3* mutant reveals stimulation of *AtHSP90* transcripts and HSP90 protein levels (5).

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
