# Peer review of "Genetically Modified Heat Shock Protein90s and Polyamine Oxidases in Arabidopsis Reveal Their Interaction under Heat Stress Affecting Polyamine Acetylation, Oxidation and Homeostasis of Reactive Oxygen Species"

_plants, 2019, doi:10.3390/plants8090323_

Round 1
Reviewer 1 Report
In their manuscript “HEAT SHOCK PROTEINS 90 and POLYAMINES 3 OXIDASES interact with the acetylation and oxidation of Polyamines” the authors want to demonstrate that HSP90 protein and PAO interact with the modification of polyamines. The assumption is mostly based on measurements of PA contents and mRNA abundance in mutant plants compared to wild type plants and inhibitor treatments.
All results for quantification of mRNA abundance are based on RT-PCR and stained gels of PCR products. In that case rather quantitative RT-PCR should be performed to see differences in mRNA abundance. In M&M it is described that both AtUBQ and 18S rDNA were used for normalization. However, in the figures only AtUBQ is shown. qRT-PCR with enough replicates and two housekeeping genes as controls are necessary to figure out whether differences in mRNA abundance are significant.
Western blot results are far below scientific standard. Actually there are no bands visible and the picture was cut to small size which does not allow judgement of specificity of the antiserum used.
Where the treatments applied under light? If yes, which intensity?
Fig 2. Two replicates (!), Significance ?
Fig 5. Statistics, significance?
Please indicate the fragment sizes that are amplified by using the primer pairs listed in table S1 and S2.
Line 69: (Spm), last bracket is missing.
Line 70-73: Please check the sentences which seem to be incomplete.
Line 422: Membranes were hybridized …. In Western blotting there is no hybridization. Please rewrite and use correct expressions for describing Western blotting. Please also specify the anti-HSP90 serum from Calbiochem, is it a plant specific one?
Line 195: Transcriptional activity of AtPAOs in the pLFY:HSP90RNAi lines revealed that AtPAOs transcripts were increased when compared to WT under control conditions. Increase of AtPAO transcripts does not reflect transcriptional activity since only mRNA abundance was tested.
Author Response
Comments and Suggestions for Authors
In their manuscript “HEAT SHOCK PROTEINS 90 and POLYAMINES 3 OXIDASES interact with the acetylation and oxidation of Polyamines” the authors want to demonstrate that HSP90 protein and PAO interact with the modification of polyamines. The assumption is mostly based on measurements of PA contents and mRNA abundance in mutant plants compared to wild type plants and inhibitor treatments.
All results for quantification of mRNA abundance are based on RT-PCR and stained gels of PCR products. In that case rather quantitative RT-PCR should be performed to see differences in mRNA abundance. In M&M it is described that both AtUBQ and 18S rDNA were used for normalization. However, in the figures only AtUBQ is shown. qRT-PCR with enough replicates and two housekeeping genes as controls are necessary to figure out whether differences in mRNA abundance are significant.
Response: Actually, qRTPCR was also performed using 18s rDNa. The relative results from the WT and the AtPao3 mutants are now shown in Fig. S4, which unfortunately was mistakenly not included amongst the submission files before. For strictly financial constrains, the other analyses were limited to RT-PCR. The results concern two replicates which nevertheless are from pooled samples from at least 4 replicates each. Thus, we are confident for the validity of the results.
Western blot results are far below scientific standard. Actually there are no bands visible and the picture was cut to small size which does not allow judgement of specificity of the antiserum used.
Response: In the Western blot we actually used 2 antibodies; the first anti PAO3 was our own produced antibody whereas the second one was a commercial one, which however exhibited low specificity to Arabidopsis HSP90. Again due to financial constrains we were not allowed to purchase the ATHSP90 antibody which has higher affinity. This is why we illustrated the relative results as histograms from image pixels treatment, along with the original Western bands, which unfortunately repeatedly revealed high background. The small sized bands were cut off to avoid the unnecessary background.
Where the treatments applied under light? If yes, which intensity?
Response: All treatments were performed under standard light intensity of 150 μmol m−2 s−1.
Fig 2. Two replicates (!), Significance ?
Response: As already mentioned, the two replicates were used for the quantification of the pixels intensity on IMAGEJ software; the germination and staining was carried in multiple independent experiments in in-vitro cultures with an average of three repetitions of petri dishes with 5 seedlings per each genotype. The results from extensive preliminary experimentation were considered as a guide regarding germination and phenotype screening (data non shown).
Fig 5. Statistics, significance?
Response: Done
Please indicate the fragment sizes that are amplified by using the primer pairs listed in table S1 and S2.
Response: Done
Line 69: (Spm), last bracket is missing.
Response: Done
Line 70-73: Please check the sentences which seem to be incomplete.
Response: Done
Line 422: Membranes were hybridized …. In Western blotting there is no hybridization. Please rewrite and use correct expressions for describing Western blotting. Please also specify the anti-HSP90 serum from Calbiochem, is it a plant specific one?
Response: The paragraphs were re-rewritten. The anti HSP90 was an antimouse one.
Line 195: Transcriptional activity of AtPAOs in the pLFY:HSP90RNAi lines revealed that AtPAOs transcripts were increased when compared to WT under control conditions. Increase of AtPAO transcripts does not reflect transcriptional activity since only mRNA abundance was tested.
Response: The sentence was re-written.
Reviewer 2 Report
This paper explores an interaction between the Heat Shock Proteins and polyamine metabolism in Arabidopsis leading to a potential model for the role of PAOs in this context. Among the recent papers on PAs in plants, this study is quite original and different. The paper is well written and concise in most aspects, and requires only a few grammatical changes – most of them are highlighted on the manuscript. A few minor comments are listed here.
The paper represents useful advances to increase our understanding of the functioning of PA metabolism and its importance in an interesting signaling pathway regulated by HSPs. The experimental design and analyses seem appropriate. A few minor comments are:
Sometime the term HSP is used as singular at other times as plural – please define up front as to whether you are using the abbreviation for one protein (HSP90) or multiple proteins – As far as I understand, HSP90 refers to a single protein. also are they all called HSP90 or HSP90s? A number of abbreviations and gene names need to be defined. Some examples are given below:
· Line 62-63 – define PhANG, SGT1, NLR etc.
· Line 187- replace “..promted to study” with “studied”
· Lines 191-194 – the font size is different…
· The statement “…resulted to…” should be replaced with “..resulted in…” at several places in the abstract and the results (Lines 24, 30, etc.
· The periods at the end of several sentences are missing; e.g. lines 52, 64, etc.
Overall, read the manuscript carefully for similar small mistakes.
Author Response
Comments and Suggestions for Authors
This paper explores an interaction between the Heat Shock Proteins and polyamine metabolism in Arabidopsis leading to a potential model for the role of PAOs in this context. Among the recent papers on PAs in plants, this study is quite original and different. The paper is well written and concise in most aspects, and requires only a few grammatical changes – most of them are highlighted on the manuscript. A few minor comments are listed here.
The paper represents useful advances to increase our understanding of the functioning of PA metabolism and its importance in an interesting signaling pathway regulated by HSPs. The experimental design and analyses seem appropriate. A few minor comments are:
Sometime the term HSP is used as singular at other times as plural – please define up front as to whether you are using the abbreviation for one protein (HSP90) or multiple proteins – As far as I understand, HSP90 refers to a single protein. also are they all called HSP90 or HSP90s?
Response: We thank the Reviewer for her/his favorable and constructive comments. Actually the Heat Shock Proteins were abbreviated with capital letters as HSP90s; the italics HSP90s refer to genes encoding for these proteins and the prefix “ At” for reference to the species Arabidopsis thaliana and the multiple alleles: AtHSP90-1, 2, 3 and 4. Same holds for polyamines oxidases (PAOs) with multiple specified proteins. When we mentioned HSP90, we refer to the HSP90 family in general.
A number of abbreviations and gene names need to be defined. Some examples are given below:
Line 62-63 – define PhANG, SGT1, NLR etc.
Response: Done
Line 187- replace “..promted to study” with “studied”
Response: Done
Lines 191-194 – the font size is different…
Response: Font size adjusted
·
The statement “…resulted to…” should be replaced with “..resulted in…” at several places in the abstract and the results (Lines 24, 30, etc.
Response: Done
The periods at the end of several sentences are missing; e.g. lines 52, 64, etc.
Overall, read the manuscript carefully for similar small mistakes.
Response: Done
Reviewer 3 Report
In this manuscript, authors described that transcriptional co-regulation between HSP90 and PAOs affects to acetylation and oxidation of PA and PA-related hydrogen peroxide homeostasis. The manuscript provides essential evidences of HSP90 for the modification of PA under various dysfunctional mutants of HSP90 and a pharmaceutical inhibitor of HSP90, radicicol. However, I suggest that several issues need to be identified and revised. My comments are below:
Major concerns,
Figures 2, 4, 5, 6 and supple Figures did not statistically analyze.
In Figure 6A, HS and HS+Rad treated WT samples are also necessary.
Authors need to show the western blot image in Figure 4C and 6B like as authors showed RT-PCR band image and its quantified graphs together.
In Figure 4A, is there any reason that authors did not analyze the AtHSP90-2 and AtHSP90-3 genes. And based on previous reports, AtHSP90-1 gene is a stress-inducible gene, while AtHSP90-2-4 genes are constitutively expressed. However, in this manuscript (Figure 4A), AtHSP90-1 is rarely detected in both control and HS (although authors determined as AtHSP90-1 is enhanced by HS), and AtHSP90-4 is specifically induced by HS, which is contradictory with previous findings.
Figure 5 data sets are not consistently designed. In Figure 5A, authors showed H2O2 contents in the excised leaves of both lines, WT and pLFY:HSP90RNAi line, however Figure 5B shows only WT, and Figure 5C does not have HS+Spd. In addition, in figure 5D, DAB staining only represented WT plants including new set of HS+Rad+guaz which is designed in Figure S3.
There is no figure legends of Supplementary Figures and Tables, especially I could not find Figure S5, which is mentioned in line 280 and 283. And I do not see any description about Fig S2 in text.
In Figure 2, two biological replicates do not seem to be enough.
In line 354-259, There is no HS data in WT plant (Figure 6). Hence, it could not be concluded as authors’ description.
Minor concerns,
There is many typos, for examples, line 52, 64, and 71, no full stop; line 85, no capital for Oxidase; no space in line 269; line 384, mn.
In line 106, 159 and 188, “HSP901-4” represents RNAi line for HSP90-1 to HSP90-4, and thus, I suggest that “HSP90-1-4” or “HSP90.1-4”.
Figure 4Bc, NATA1 data needs to show the control image, AtUBQ.
Figure 3, 4Bb, 4Bc, 4C, 5, 6B, and S3Bb need to be identified as “underbar” for each plant, such as WT and pLFY:HSP90RNAi.
Figure 6A right graph, “AtHSSP90-4” changes to “AtHSP90-4”.
Figure S3Bc needs to be quantified.
In line 379, does 8/16h photoperiod mean 8 h light and 16 h dark?
Author Response
Review Report Form
English language and style
( ) Extensive editing of English language and style required
( ) Moderate English changes required
(x) English language and style are fine/minor spell check required
( ) I don't feel qualified to judge about the English language and style
Response: English was improved.
Yes | Can be improved | Must be improved | Not applicable | |
Does the introduction provide sufficient background and include all relevant references? | (x) | ( ) | ( ) | ( ) |
Is the research design appropriate? | (x) | ( ) | ( ) | ( ) |
Are the methods adequately described? | (x) | ( ) | ( ) | ( ) |
Are the results clearly presented? | ( ) | (x) | ( ) | ( ) |
Are the conclusions supported by the results? | ( ) | (x) | ( ) | ( ) |
Comments and Suggestions for Authors
In this manuscript, authors described that transcriptional co-regulation between HSP90 and PAOs affects to acetylation and oxidation of PA and PA-related hydrogen peroxide homeostasis. The manuscript provides essential evidences of HSP90 for the modification of PA under various dysfunctional mutants of HSP90 and a pharmaceutical inhibitor of HSP90, radicicol. However, I suggest that several issues need to be identified and revised. My comments are below:
Major concerns, Figures 2, 4, 5, 6 and supple Figures did not statistically analyze.
Response: We thank the Reviewer for her/his favorable and constructive comments. In the revised manuscript we have completed the statistics that was indeed missing from the original submission in Fig. 5, since the data represent average values from independent repetitions. We qualified futile to present statistical support in the pixels-quantified bands (in the rest of the Figures) since the histograms represent a numeric exploitation of the gels bands for more efficient detection of gels results and for minimizing visual incertitude when checking the bands.
In Figure 6A, HS and HS+Rad treated WT samples are also necessary.
Response: This result regarding the WT HS and HS+Rad is already presented in Fig. 4Aa. We have avoided to present the same result just to point up a direct comparison between WT and Atpao3 profiles, because it will be presentation of same result twice.
Authors need to show the western blot image in Figure 4C and 6B like as authors showed RT-PCR band image and its quantified graphs together.
Response: We do not show the Western blot results directly in the principal Figure because of the permanent issue of weak quality that we encountered with the immuno detection despite the numerous repetitions and efforts, as already mentioned. Instead, we present the band intensities as histograms in the S-Figures.
In Figure 4A, is there any reason that authors did not analyze the AtHSP90-2 and AtHSP90-3 genes.
And based on previous reports, AtHSP90-1 gene is a stress-inducible gene, while AtHSP90-2-4 genes are constitutively expressed. However, in this manuscript (Figure 4A), AtHSP90-1 is rarely detected in both control and HS (although authors determined as AtHSP90-1 is enhanced by HS), and AtHSP90-4 is specifically induced by HS, which is contradictory with previous findings
Response: Actually, the results presented in this manuscript is a part of a large study on the potential interaction(s) between the AtHSP90 cytosolic gene family (1, 2, 3 and 4) and the AtPAOs (1, 2, 3, 4 and 5). To simplify the presentation of the results, we selected to include in this manuscript the most obvious and contrasting results of the project, that is the two genes AtHSP90-1 and 4, taking into account that the AtHSP90-2-3 and 4 are regarded as constitutively expressed and the AtHSP90-1, as stress-inducible. Indeed the AtHSP90-1 was largely shown throughout the entire project to be a principally stress responsive gene; interestingly however, the AtHS90-1 was also expressed under physiological conditions with a trace amount, compared to the other genes. Nevertheless both genes were induced by HS and not only AtHSP90-4. The AtHSP90-1 was more inhibited when using radicicol, which confirms the established finding, meaning that AtHSP90-1 is highly qualitatively HS-inducible.
Figure 5 data sets are not consistently designed. In Figure 5A, authors showed H2O2 contents in the excised leaves of both lines, WT and pLFY:HSP90RNAi line, however Figure 5B shows only WT, and Figure 5C does not have HS+Spd. In addition, in figure 5D, DAB staining only represented WT plants including new set of HS+Rad+guaz which is designed in Figure S3.
Response: Actually the legend is explaining the aim of the Figure, which is not the endogenous production of H2O2 itself but rather the PAOs activity that is assessed by H2O2 as a specific PAOs' activity product. In other terms, the main result of Fig. 5 is presented in the part A, representing the stimulation of PAOs activity (analogous to H2O2 production) via HS and moreover via inhibition of HSP90s. Part B further confirms these data. Fig. 5B shows (independently of the genotype) that the increase of H2O2 following the treatments is indeed produced via the specific Spd oxidizing reaction by PAOs. In Fig. 5C we determined the endogenous H2O2 produced in the relevant samples, as an expected consequence of PAOs activity. Here we quantified the H2O2 produced as a ROS species. Actually production of H2O2 by PAO has been repeatedly confirmed in the literature. Herein we aimed to illustrate the double inhibitory effect of Rad and guazatine on the overall studied mechanism.
There is no figure legends of Supplementary Figures and Tables, especially I could not find Figure S5, which is mentioned in line 280 and 283. And I do not see any description about Fig S2 in text.
Response: We apologize, as an error occured during the submission process. We now include the legend of S-Figures as well as the Fig S2 which illustrates the semiQRT-PCR results.
In Figure 2, two biological replicates do not seem to be enough.
Response: As already mentioned. the two replicates were used for the quantification of the pixels intensity on IMAGEJ software. Germination and staining was carried on multiple independent experiments in in-vitro cultures with an average of at least three repetitions of petri dishes with 5 seedlings per each genotype. This experiment result was one exploitation amongst a general experiment that concerned germination and phenotype screening (data not shown).
In line 354-259, There is no HS data in WT plant (Figure 6). Hence, it could not be concluded as authors’ description.
Response: Done.
Minor concerns,
There is many typos, for examples, line 52, 64, and 71, no full stop; line 85, no capital for Oxidase; no space in line 269; line 384, mn.
In line 106, 159 and 188, “HSP901-4” represents RNAi line for HSP90-1 to HSP90-4, and thus, I suggest that “HSP90-1-4” or “HSP90.1-4”.
Response: Done
Figure 4Bc, NATA1 data needs to show the control image, AtUBQ.
Response: The normalization of NATA1 bands was realized according the AtUBQ (Fig4Ba) as the results Fig4Ba and Fig 4Bb were obtained from the same experimental set.
Figure 3, 4Bb, 4Bc, 4C, 5, 6B, and S3Bb need to be identified as “underbar” for each plant, such as WT and pLFY:HSP90RNAi.
Response: Figures adjusted.
Figure 6A right graph, “AtHSSP90-4” changes to “AtHSP90-4”.
Response: Done
Figure S3Bc needs to be quantified.
Response: Done
In line 379, does 8/16h photoperiod mean 8 h light and 16 h dark?
Response: Yes
Round 2
Reviewer 1 Report
I still think that all the gene expression analyses should be done by qRT-PCR rather than by stained agarose gels. Also, there is no direct evidence that "HEAT SHOCK PROTEIN90 and POLYAMINE OXIDASE interact...." as written in the title. The result of immunoblot analysis is not suitable to draw any conclusion. Therefore, my judgement for this manuscript: reject.
Author Response
Dear Reviewer
We are respectful of your decision